# Lanthanum Exchanged Keggin Structured Heteropoly Compounds for Biodiesel Production

**Badriah Al-Shammari, Qana A. Alsulami**  **and Katabathini Narasimharao ***

Department of Chemistry, Faculty of Science, King Abdulaziz University, P.O. Box 80203, Jeddah 21589, Saudi Arabia; badriah.alshammari@gmail.com (B.A.-S.); qalselami@kau.edu.sa (Q.A.A.)
* Correspondence: nkatabathini@kau.edu.sa; Tel.: +966-538638994; Fax: +966-26952292

**Abstract:** La-exchanged 12-tungstophosphoric acid ($La_x$TPA) and 12-molybdophosphoric acid ($La_x$MPA) salts (x = 0.25, 0.50, 0.75 and 1.00) were prepared via an ion exchange method. The physico-chemical characteristics of the materials were analyzed by using elemental analysis, X-ray diffraction (XRD), Fourier transformed infrared (FT-IR) spectroscopy, scanning electron microscopy (SEM), $N_2$-physical adsorption, X-ray photoelectron spectroscopy (XPS), and acidity-basicity measurements. The results indicated that La was introduced into the secondary structure of heteropolyacid (HPA) and have not influenced the primary structure, which effectively improved the surface area and pore size. Acidity-basicity studies indicated that incorporation of La resulted in a decrease in the number of acid sites and an increase in the number of basic sites. The catalytic activity of samples was studied in transesterification of glyceryl tributyrate with methanol and $La_x$TPA samples which exhibited high activity compared to $La_x$MPA samples due to having more active basic sites and a larger surface area. Calcined $La_x$TPA samples showed excellent stability, outstanding recyclability, and high activity for one pot transesterification and esterification processes. This outcome was attributed to the presence of balanced acidic and basic sites.

**Keywords:** $H_3PW_{12}O_{40}$; $H_3PMo_{12}O_{40}$: La exchange; biodiesel; transesterification; esterification

---

## 1. Introduction

The growth in demand for conventional petroleum based fuels, particularly diesel, has resulted in enormous concern regarding increased greenhouse gas emissions from the consumption of diesel [1,2]. It is well known that the emission of greenhouse gases is responsible for global warming [3]. In this context, researchers have been exploring different ways to develop sustainable and renewable alternative fuels or fuel additives for partial or total replacement of petroleum diesel. Biodiesel is considered as a renewable fuel and can be utilized in cycle engines and also in stationary motors [4]. Biodiesel could be obtained by a chemical reaction between different triglycerides presented in vegetable oils, animal fats, and alcohol [5]. Production of methyl esters (biodiesel) by transesterification involves a triglyceride reaction with methanol in presence of a catalyst, forming biodiesel [6].

Acidic catalysts have an edge in the production of biodiesel due to providing both esterification of fatty acids and transesterification of triglycerides processes. Vieira et al. [7] evaluated the use of $La_2O_3$, sulfated $La_2O_3$, HZSM-5, and $SO_4^{2-}/La_2O_3/$HZSM-5 as heterogeneous catalysts for biodiesel production. The heteropolyacids (HPAs) catalysts are considered to be economically feasible catalysts for biodiesel production, and they showed an ability to promote the esterification and transesterification process more efficiently than $H_2SO_4$ [8]. Cao et al. [9] used an $H_3PW_{12}O_{40}$ $6H_2O$ catalyst for transesterification of waste cooking oil. In the presence of free fatty acids, $H_3PW_{12}O_{40}/Ta_2O_5$ exhibited superior performance for both esterification and transesterification [10].

Cesium salts of TPA were used for transesterification of vegetable oil and superior catalytic performance was observed as shown in the case of homogeneous catalysts (NaOH or $H_2SO_4$) with additional advantages of easy separation of the catalyst from the reaction mixture and its reuse [11]. Narasimharao et al. [12] synthesized $Cs_xH_{3-x}PW_{12}O_{40}$ ($x = 0.9$–3.0) salts and the authors observed that Cs salts of TPA are active catalysts for esterification of palmitic acid and transesterification of glyceryl tributyrate, which are essential reactions for biodiesel production. It was observed that incorporation of metal cations such as $Cs^+$ in place of $H^+$ cations resulted in interesting effects on the surface area, pore structure, and solubility. These properties are significant for any catalyst, as they play a role in the resistance against deactivation through solvents and thus improve their recycling potential [13]. It is worth mentioning that employing heterogeneous catalysts for biodiesel production based on HPAs is timely and extremely important, because these catalysts can be used in industry [14] for greener fuels [15] with easy separation between the target (biodiesel) and byproducts (mainly glycerol). Zhao et al. [16] synthesized La-modified 12-molybdophosphoric acid samples with various La/Mo ratios and utilized them as catalysts for a fructose to lactic acid reaction. Li et al. [17] synthesized $La^{3+}/H_3PW_{12}O_{40}$ material through the hydrothermal method and used it as a photocatalyst for effectual solar light-driven photocatalytic degradation of methyl orange and rhodamine B dyes. However, there is no report in the current literature on the application of La exchanged HPAs as catalysts for biodiesel production.

In the present work, we synthesized La-exchanged 12-tungstophosphoric acid and 12-molybdophosphoric acid catalysts. We then effectively used them for transesterification of glyceryl tributyrate with methanol. The influence of reaction conditions over the performance of catalysts was also studied by changing the reaction time, temperature, and catalyst loading. The synthesized materials were analyzed by using Inductive coupled plasma-atomic emission spectroscopy (ICP-AES), X-ray diffraction (XRD), Scanning electron microscopy (SEM), $N_2$-physisorption, X-ray photoelectron spectroscopy (XPS), and acid-base properties with Fourier transformed infrared (FT-IR) spectroscopy techniques. The correlation between the catalytic performance and physico-chemical characteristics of the catalysts was also studied.

## 2. Results and Discussion

The phase composition and crystal structure of the synthesized materials were studied using a powder XRD technique. The XRD patterns of both $La_x$TPA and $La_x$MPA samples are shown in Figure 1. The pure TPA sample, which was calcined at 300 °C, showed all the reflections corresponding to a cubic Pn3m space group [JCPDS PDF#50-0657]. The XRD pattern of $La_{0.25}$TPA sample exhibited a new set of sharp reflections. The reflections corresponding to pure TPA disappeared as the La atom composition increased beyond 0.50 ($La_{0.75}$TPA and $La_{1.0}$TPA samples). The shift in 2θ positions of reflections towards lower angles in the $La_x$TPA samples is consistent with the previously reported results [16,17]. The XRD patterns of calcined MPA and $La_x$MPA samples are also shown in Figure 1.

The reflections of the calcined MPA sample are similar to the characteristics of the monoclinic Keggin structure. After incorporating the La, the reflections due to MPA disappeared and new reflections due to lanthanum salt of MPA appeared in all La exchanged MPA samples. It is interesting to note that the XRD patterns of $La_x$MPA samples also exhibited few broad humps due to the presence of an unknown amorphous material, which was not observed in the case of $La_x$TPA samples. This is probably due to the fact that MPA thermal stability is lower than TPA and a slight amount of Keggin compound was decomposed into oxides when $La_x$MPA samples were used. The sharp reflections observed in all the materials revealed that synthesized $La_x$TPA and $La_x$MPA are highly crystalline. The most intense XRD reflections for bulk TPA, MPA, and La exchanged salts were used to determine the crystallite size by applying the Scherrer equation, and the results are presented in Table 1. The results show that the crystallite size decreased from 50 nm to 20 nm as the La composition increased to $La_{0.25}$TPA, and the crystallite size increased to 25 nm for $La_{1.0}$TPA material, demonstrating the formation of agglomerates in the case of $La_{1.0}$TPA.

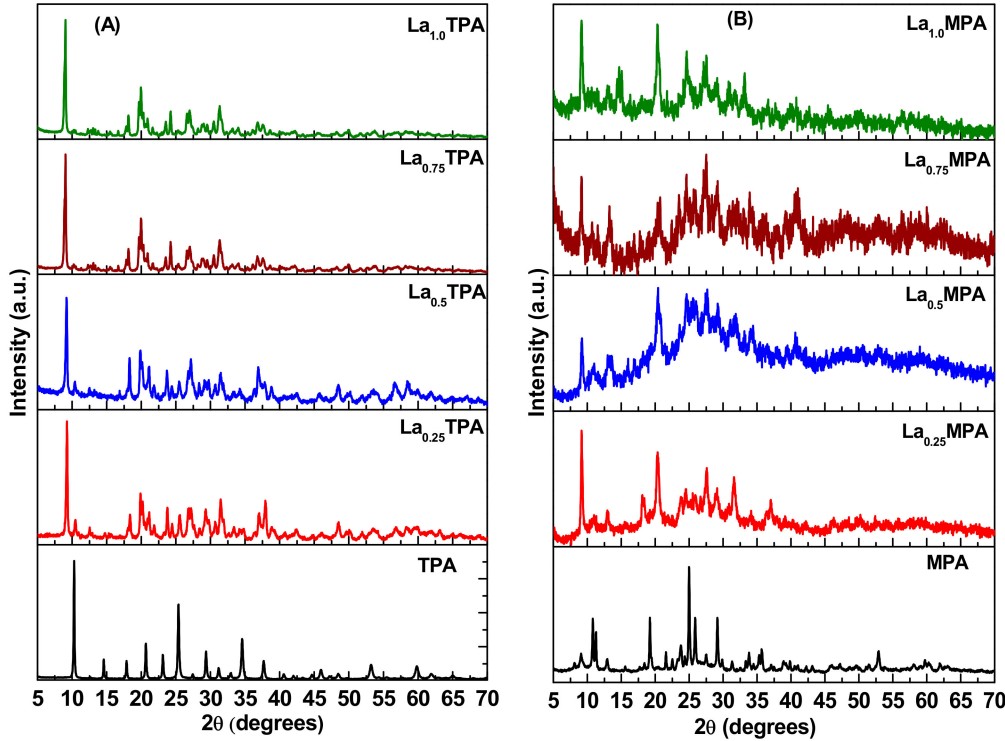

**Figure 1.** XRD patterns of (**A**) $La_x$TPA and (**B**) $La_x$MPA samples

**Table 1.** Data obtained from XRD and $N_2$-physisorption measurements.

| Sample | Crystallite Size (nm) | BET Surface Area ($m^2g^{-1}$) | Average Pore Width (Å) | Average Pore Volume ($cm^3g^{-1}$) |
|---|---|---|---|---|
| $La_{0.25}$TPA | 110 | 8.5 | 50 | 0.009 |
| $La_{0.50}$TPA | 125 | 12.3 | 51 | 0.023 |
| $La_{0.75}$TPA | 137 | 16.4 | 54 | 0.054 |
| $La_{1.00}$TPA | 150 | 21.4 | 62 | 0.075 |
| $La_{0.25}$MPA | 90 | 6.2 | 48 | 0.008 |
| $La_{0.50}$MPA | 103 | 10.5 | 50 | 0.020 |
| $La_{0.75}$MPA | 115 | 13.0 | 56 | 0.050 |
| $La_{1.00}$MPA | 120 | 15.8 | 60 | 0.070 |

To study the structural features of synthesized samples, FT-IR analysis was also performed. The FT-IR spectra of $La_x$TPA and $La_x$MPA samples are shown in Figure 2. The FT-IR spectrum of the TPA sample showed Keggin structure characteristic bands at 1080 $cm^{-1}$ (P-$O_i$), 965 $cm^{-1}$ (W-$O_t$), 885 $cm^{-1}$ (W-$O_c$-W), and 755 $cm^{-1}$ (W-$O_e$-W), where $i$, $t$, $c$, and $e$ represent the precise position of oxygen atoms (internal, terminal, corner, and edge-shared) in the Keggin structure [12]. It is clear that all the $La_x$TPA samples exhibited characteristic bands of the Keggin structure, which is an indication that the original structure was intact after the La exchange. However, it is interesting to note that position of the band corresponding to W-$O_e$-W stretching mode shifted from 755 $cm^{-1}$ to 782 $cm^{-1}$, 786 $cm^{-1}$, 790 $cm^{-1}$, and 800 $cm^{-1}$ for $La_{0.25}$TPA, $La_{0.5}$TPA, $La_{0.75}$TPA, and $La_{1.0}$TPA, respectively.

The Keggin ion in pure calcined MPA sample shows FT-IR bands at 1060 $cm^{-1}$, 960 $cm^{-1}$, 870 $cm^{-1}$, and 750 $cm^{-1}$ corresponds to P-$O_i$, Mo-$O_t$, Mo-$O_c$-Mo, and Mo-$O_e$-Mo bonds, respectively [18]. All the FT-IR bands corresponding to Keggin ion appeared in $La_x$MPA samples, and also a shift in the peak position of Mo-$O_e$-Mo stretching mode was observed in these samples. The interaction between the Keggin anion $[PW_{12}O_{40}]^{3-}$ or $[PMo_{12}O_{40}]^{3-}$ and the Lewis acid $La^{3+}$ ions forms La-O-W bonds. This synergistic effect could change some characteristic peak positions. However, it is important to understand the location of La ions. It is well known that incorporated foreign metal ion in the Keggin

structure could induce variations in the FT-IR spectra of samples [19]. It was previously reported that incorporated metal ion reduced the symmetry of $[PO_4]^{3-}$ ion, which resulted in a split in the P-O stretching band [18]. The FT-IR spectra of $La_xTPA$ and $La_xMPA$ materials did not exhibit any peaks at 1080 cm$^{-1}$ or 1060 cm$^{-1}$ after La exchange. These results revealed that La atoms did not replace the W or Mo in peripheral positions of the Keggin ion, but they do act as cations in the secondary structure.

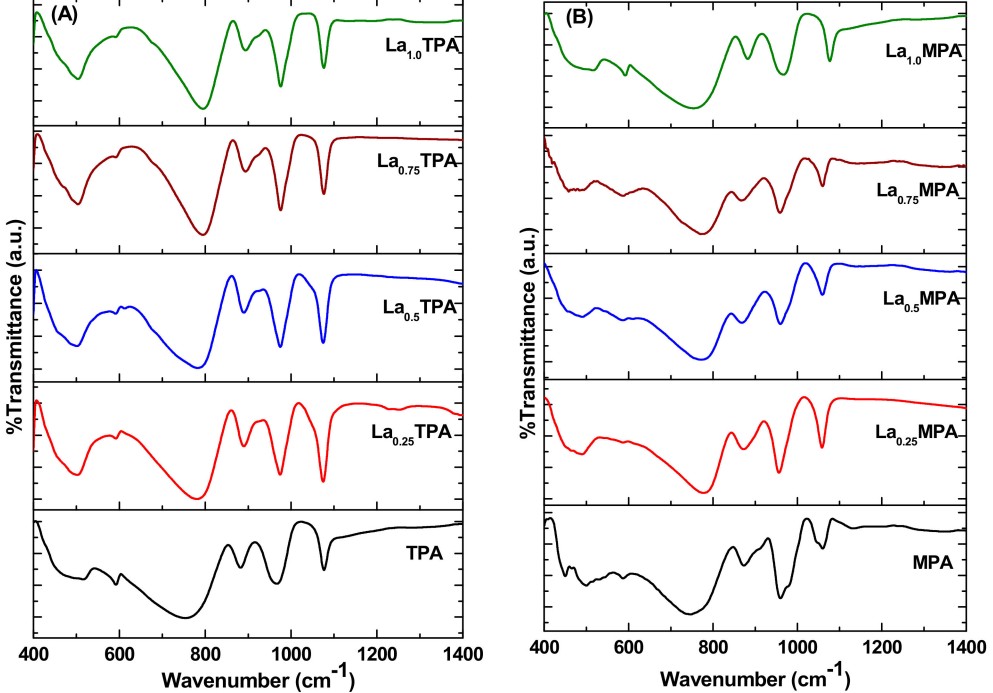

**Figure 2.** FT-IR spectra of (**A**) $La_xTPA$ and (**B**) $LA_xMPA$ samples

The structural aspects of the synthesized La exchanged TPA and MPA samples were investigated further with the DR UV-vis technique. The DR UV-vis spectra of $La_xTPA$ and $La_xMPA$ materials are presented in Figure 3. The edge energy values were determined from the wavelength of the UV-vis absorption edge. The pure TPA and MPA samples clearly showed a representative weak absorption band at 260 nm (4.76 eV), which could be assigned to an O-W charge transfer in the Keggin ion [20]. A wide peak centered at 365 nm (3.39 eV) was also observed which was due to the ligand-metal charge transfer from O to $W^{6+}$ in the Keggin anion; these observations are in agreement with a previous research report [17]. The $La_xTPA$ samples exhibited similar absorption peaks to that of the bulk TPA sample. However, the adsorption edge for $La_{1.00}TPA$ samples was shifted to 3.54 eV, indicating that incorporation of La ions resulted in the shifting of the edge energy towards the visible-light region. Also, an increase of the La content resulted in an increase in the peak intensity, indicating that La concentration affected the optical absorption properties. The $La_{1.00}TPA$ sample showed better optical absorption ability compared to $La_{0.25}TPA$ and other synthesized samples. The DR UV-vis absorption spectra of MPA and $La_xMPA$ samples were similar to that of TPA and $La_xTPA$ samples. The wide absorption peak (due to LMCT) wavelength value of MPA and $La_xMPA$ was higher than for TPA and $La_xTPA$ samples. It was observed that the bulk MPA exhibited an energy edge at 2.47 eV which was shifted to 2.75 eV for the $La_xMPA$ samples. These results indicated that the $La_xMPA$ samples possessed an energy edge in the visible light region.

The morphology and microstructure of the investigated samples were studied by SEM analysis. Figure S1 exhibits the SEM micrographs of investigated samples. The SEM images of TPA and $La_{0.25}TPA$ samples revealed that these samples consisted of nanoparticles with an average pore width of around 35 nm. An increase of the La content resulted in an increase of particle size. It was observed that the large size irregular particles were formed by the aggregation of smaller particles. The largest size

particles (with an average particle size of 115 nm) were observed for the $La_{1.00}TPA$ sample. The XRD results are in good agreement that the reflection that La exchanged TPA samples are sharper compared to the TPA sample. The SEM micrographs of MPA and $La_xMPA$ samples revealed that these samples are composed of spherical particles. Large-sized particles were observed in the case of $La_xMPA$ samples which were similar to that of $La_xTPA$ samples.

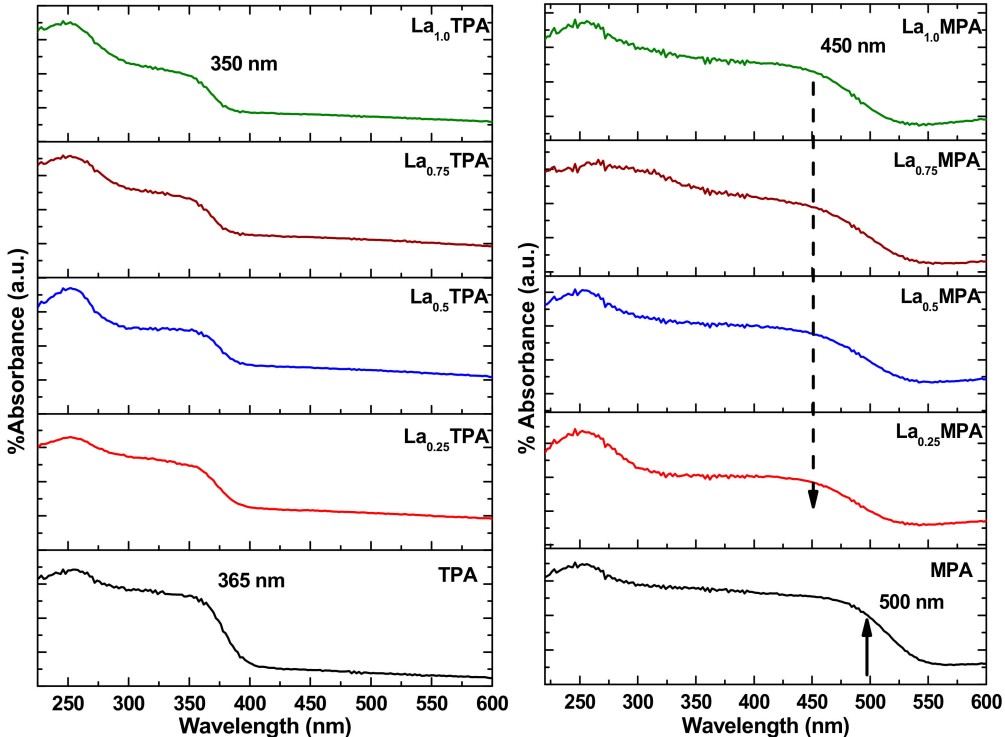

**Figure 3.** DR UV-vis spectra of $La_xTPA$ and $La_xMPA$ samples.

It is clear that the amount of La had a significant influence on the HPA particle size, however similar morphology of $La_xTPA$ and $La_xMPA$ samples indicated that the La content did not change the morphology of the samples. The SEM analysis results indicated that $La_xTPA$ and $La_xMPA$ materials were formed due to the self-assembly of several smaller HPA particles. The textural characteristics of the $La_xTPA$ and $La_xMPA$ samples were obtained from the $N_2$ physisorption results. The nitrogen adsorption-desorption isotherms of the investigated samples are depicted in Figure S2. The nitrogen adsorption-desorption isotherms of TPA and $La_xTPA$ samples are similar, and both show a slight increase with any increases of $P/P^0$. The isotherms clearly belong to Type II isotherms [21], which are generally due to a single to multi-layer reversible adsorption process occurring on non-porous or macroporous solid surfaces. The observed results indicate that the samples possessed macro size pores, consistent with the measurement of the pore size.

Similar results were observed in the case of MPA and $La_xMPA$ samples, as shown in Figure S2 in the Supplementary Materials. The $S_{BET}$ values increased from 2.5 $m^2g^{-1}$ (for TPA) to 36 $m^2g^{-1}$ (for the $La_{1.00}TPA$ sample). The observed results are in accordance with morphological changes of the samples, and previous reports also reported similar observations [22] suggesting that metal salts of TPA have higher surface areas compared to TPA. The pore size distribution measurements revealed an increase in the average pore width from 50 Å to 62 Å with an increase of the La amount (Table 1). It was reported that large voids exist among the micro crystallites of the metal exchanged HPA salts [23], as closely packed particles aggregates could form mesoporous voids. Inter particle spaces between micro crystallites are responsible for the increase in the average pore width for the La exchanged TPA and MPA samples.

Figure 4 represents the deconvoluted XP spectra of representative samples. The La *3d* photoelectron spectra for $La_{0.25}TPA$, $La_{1.00}TPA$, $La_{0.25}MPA$, and $La_{1.00}MPA$ samples can be seen in Figure 6. It was reported that La *3d* core spectrum splits into $3d_{5/2}$ and $3d_{3/2}$ lines because of spin-orbit interaction. The La exchanged HPA samples showed XPS peaks at around 836.6 eV, 839.7 eV, 853.3 eV, and 856.3 eV related to La $3d_{5/2}$ and La $3d_{3/2}$ components, respectively. And the two peaks in each line could be assigned to main and satellite peaks due to $3d^0 4f^0$ and $3d^0 4f^1L$ configurations [24]. It was observed that different types of La compounds exhibit satellite peaks at different binding energies [25]. It was previously observed that the pure $La_2O_3$ displayed peaks at 835.8 eV and 839.3 eV related to La $3d_{5/2}$ components ($La^{+3}$ species) [26]. From the XP spectra, it is clear that La $3d_{5/2}$ and La $3d_{3/2}$ peaks were observed at higher binding energy values, which is mainly due to the strong interaction of $La^{3+}$ with O moieties from the Keggin anion.

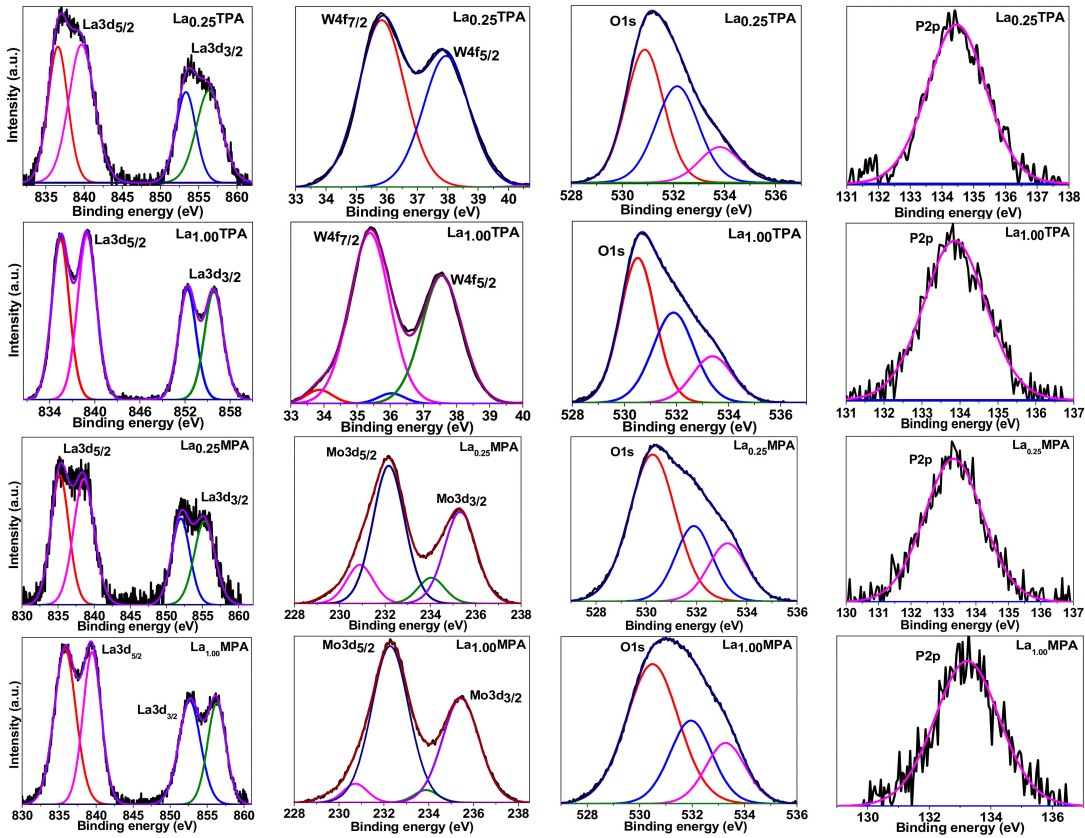

**Figure 4.** Deconvoluted X-ray photoelectron spectra for $La_{0.25}TPA$, $La_{1.00}TPA$, $La_{0.25}MPA$, and $La_{1.00}MPA$ samples.

The W*4f* XP peaks for $La_{0.25}TPA$ sample appeared at 35.8 eV and 37.9 eV and were related to W $4f_{7/2}$ and W $4f_{5/2}$ contributions. It was reported that pure TPA sample ($W^{6+}$) exhibits W$4f_{7/2}$ and W$4f_{5/2}$ peaks at 35.8 eV and 37.8 eV [27], respectively. Therefore, the surface characteristics of $La_{0.25}TPA$ could be very similar to the TPA sample, as a very small amount of La was incorporated into the HPA structure. On other hand, the $La_{1.00}TPA$ sample exhibited two different W$4f_{7/2}$ and W$4f_{5/2}$ contributions corresponding to $W^{6+}$ and $W^{5+}$ species (33.9 eV and 36.0 eV). This mainly due to the presence of some oxygen defect sites in $La_{1.00}TPA$, so some part of $W^{6+}$ is reduced to $W^{5+}$. It was observed that both $La_{0.25}MPA$ and $La_{1.00}MPA$ materials show two sets of Mo$3d_{5/2}$ and Mo$3d_{3/2}$ species corresponding to $Mo^{6+}$ and $Mo^{5+}$, while Mo$3d_{5/2}$ peaks observed at 232.2 eV and 231.0 eV can be attributed to $Mo^{6+}$ and $Mo^{5+}$ species of the HPA [28]. It is well known that Mo based HPAs are more easily reducible than the W based HPAs, while electron localization occurs in a reduced Mo or W Keggin structure.

A single broad P *2p* XP peak was observed in case of all the investigated La exchanged HPA samples. It was previously reported that in a well crystallized HPA structure consists of a single XP peak between 133–134 eV [28] and also the FWHM of peaks appears to be almost the same. These observations suggest that there is no collapse of the Keggin structure of the HPA. The XP spectra of the samples showed three O *1s* peaks at 531.2 eV, 532.6 eV, and 534.1 eV, respectively. The peak at 531.2 eV could be assigned to W-O-W or Mo-O-Mo species [29]. The peak at 532.6 eV is attributed to W-O-P or Mo-O-P species. The third peak at 534 eV could be attributed to O-La species on the surface of the sample. There is a clear possibility for the presence of water molecules to seen on the sample surface, however these samples are calcined at 300 °C and a previous XPS study revealed that the binding energy of O *1s* in water appears at around 535.5 eV [30].

The bulk and surface composition of $La_x$TPA and $La_x$MPA materials were studied by ICP-AES and XPS techniques, respectively. A good agreement between theoretical and actual La concentration was observed across the investigated range $La_{0.25}$–$La_{1.00}$ (Table 2). The surface La concentration is marginally lesser than the bulk concentration, indicating that the surface of catalyst has La-depletion. It was observed that there is a considerable difference between the surface and bulk La/W(Mo) ratios in the case of $La_{0.25}$TPA and $La_{0.25}$MPA samples. The observed deviation could be due to the formation of La exchanged salt, during which $La_xPW_{12}O_{40}$/$La_xPMo_{12}O_{40}$ crystals were covered by a surface layer of TPA or MPA, while a similar phenomenon was observed in the case of potassium salt from the heteropolyacid catalyst [31]. In order to understand the type of acidic sites presented in the investigated catalysts, the samples were analyzed by the pyridine infrared adsorption method, and the results are shown in Figure 5. It is well known that there are three major bands with respect to pyridine adsorption over the surface acid sites. The band that appeared at 1450 $cm^{-1}$ was due to the coordinately bonded pyridine over the Lewis (L) acid sites and the band that appeared at 1545 $cm^{-1}$ was mainly caused by pyridinium ion over the Brönsted (B) acid sites. The third IR band that appeared at 1505 $cm^{-1}$ was due to the pyridine molecules that bonded with both L and B acid sites [32]. From Figure 5, it is clear that $La_x$TPA and $La_x$MPA materials possessed both B and L acid sites. The B acid sites of the samples are created by the dissociation of $H_2O$ molecules over $La^{3+}$ HPA structure; $[La(H_2O)_n]^{3+}[PW_{12}O_{40}]$ or $[La(OH)(H_2O)_{n-1}]^{2+}H^+[PW_{12}O_{40}]$.

**Table 2.** Bulk and surface elemental composition of La exchanged TPA and MPA samples.

| Catalyst | Bulk Composition (Atom %) | | | | | | | Surface Composition (Atom %) | | | | |
|---|---|---|---|---|---|---|---|---|---|---|---|---|
| | La | W | Mo | O | P | La/W | La/Mo | La | W | Mo | O | P |
| $La_{0.25}$TPA | 1.9 | 18.2 | - | 77.6 | 2.3 | 0.10 | - | 1.8 | 16.3 | - | 79.7 | 2.1 |
| $La_{0.50}$TPA | 3.8 | 16.4 | | 77.8 | 2.0 | 0.23 | - | 3.6 | 15.4 | - | 79.1 | 1.9 |
| $La_{0.75}$TPA | 6.5 | 15.1 | - | 76.6 | 1.8 | 0.43 | - | 6.1 | 14.5 | - | 77.5 | 1.7 |
| $La_{1.00}$TPA | 8.6 | 13.6 | - | 75.9 | 1.9 | 0.63 | - | 8.2 | 12.6 | - | 77.5 | 1.7 |
| $La_{0.25}$MPA | 1.8 | - | 17.1 | 79.3 | 1.8 | - | 0.10 | 1.4 | - | 15.2 | 81.6 | 1.6 |
| $La_{0.50}$MPA | 3.9 | - | 16.2 | 77.9 | 2.0 | - | 0.24 | 3.4 | - | 14.4 | 80.3 | 1.9 |
| $La_{0.75}$MPA | 7.0 | - | 13.5 | 77.5 | 2.0 | - | 0.51 | 6.4 | - | 12.8 | 78.8 | 1.9 |
| $La_{1.00}$MPA | 8.1 | - | 13.0 | 77.1 | 1.8 | - | 0.62 | 6.9 | - | 12.0 | 79.4 | 1.5 |

The pure TPA and MPA possess the largest number of acid sites compared to the La exchanged samples and increase of La content led to a decrease in both B and L acid sites. The reason for the differences in the total number of acid sites of $La_x$TPA and $La_x$MPA samples and bulk HPAs was that the La exchanged samples possessed extra L acid sites due to the presence of $La^{3+}$ and also unsaturated $Mo^{6+}$ or $W^{6+}$ cations, whereas the B acid sites had unexchanged protons [33]. To evaluate the basic properties of the La exchanged TPA and MPA samples, the H-donor pyrrole was used as a reacting molecule. The main features of FT-IR spectra of pyrrole adsorbed samples are presented in Figure 5. Researchers utilized the ν(NH) stretching frequencies in the NH-O hydrogen bridge to measure the basic strength of zeolites and metal oxides [32]. It was previously reported that the pyrrole adsorbed samples exhibits different types of narrow and broad bands due to the pyrrole adsorbed on basic

sites [34]. The La exchanged TPA and MPA samples exhibited five different new bands at 3678 cm$^{-1}$ (small), 3470 cm$^{-1}$ (broad), 1620 cm$^{-1}$ (sharp), 1450 cm$^{-1}$ (small), and 1375 cm$^{-1}$ (small) compared to bulk TPA and MPA samples. It was well reported that the several peaks appear in the range of 3560–2850 cm$^{-1}$ as combination bands, which are associated with strong basic sites [35]. The bands that appeared at 1630 cm$^{-1}$ and 1450 cm$^{-1}$ are very similar for liquid pyrrole and are suggestive of pyrrole being chemisorbed on basic sites [36]. The results from the figure clearly indicated that the intensity of the peaks due to basic sites increased with an increase of the La content in the HPA structure. The total acidic and basic sites per unit surface area values were calculated and presented in Table S1. As expected, the La$_{1.0}$TPA sample possessed a large number of basic sites per unit surface area.

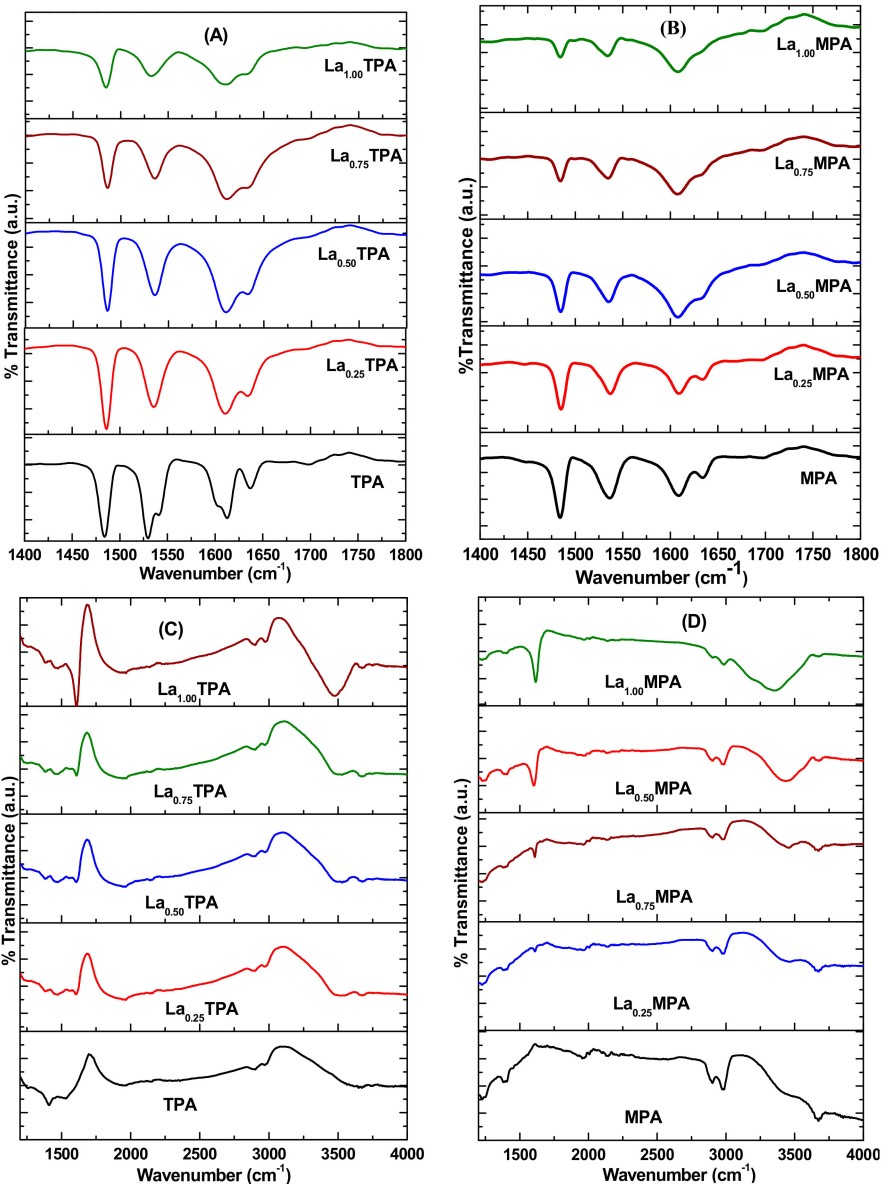

**Figure 5.** Acidity and basicity measurements using probe molecules and FTIR spectroscopy: (**A**,**B**)-pyridine adsorbed samples, (**C**,**D**)-pyrrole adsorbed samples.

Transesterification of glyceryl tributyrate with methanol is a reversible chemical reaction to produce methyl-esters (also known as biodiesel) and glycerin, and this process comprises three different elementary reactions [12]. In the first elementary step, a reaction between triglycerides and methanol yields methyl-esters and diglycerides. In the subsequent step [Equation (1)], the reaction

between diglycerides and methanol obtains methyl-esters and monoglycerides. In the last step, the monoglycerides turn into methyl-esters and glycerin.

$$\text{Triglyceride + Methanol} \rightarrow \text{Diglyceride + Methyl ester} \tag{1}$$

$$\text{Diglyceride + Methanol} \rightarrow \text{Monoglyceride + Methyl ester} \tag{2}$$

$$\text{Monoglyceride + Methanol} \rightarrow \text{Glycerol + Methyl ester} \tag{3}$$

Catalytic transesterification activity measurements over $La_xTPA$ and $La_xMPA$ materials were performed to obtain the proper reaction conditions by changing different parameters such as the reaction temperature, reaction time, amount of catalyst, and the methanol/glyceryl tributyrate molar ratio. To evaluate the effect of $La^{3+}$ ions in HPA structure on the transesterification activity, both $La_xTPA$ and $La_xMPA$ samples were tested for their reactions.

Figure 6 represents the conversion of glyceryl tributyrate and selectivities of different products obtained at three different reaction temperatures (70, 80, and 90 °C) using $La_xTPA$ and $La_xMPA$ materials. The obtained results clearly indicated that $La_xTPA$ samples are efficient catalysts for transesterification compared to $La_xMPA$ samples. The conversion of glyceryl tributyrates were enhanced with an increase of the reaction temperature in all catalyst samples. The $La_{1.00}TPA$ sample exhibited 90.5%, 94.2%, and 98.1% of glyceryl tributyrate conversion at 70 °C, 80 °C, and 90 °C, respectively, after 150 min of reaction time. On other hand, the $La_{1.00}MPA$ sample exhibited 74.8%, 82.5%, and 86.9% under same reaction conditions. The selectivities of transesterification products were also changed when the reaction temperature was altered for both series of $La_xTPA$ and $La_xMPA$ samples.

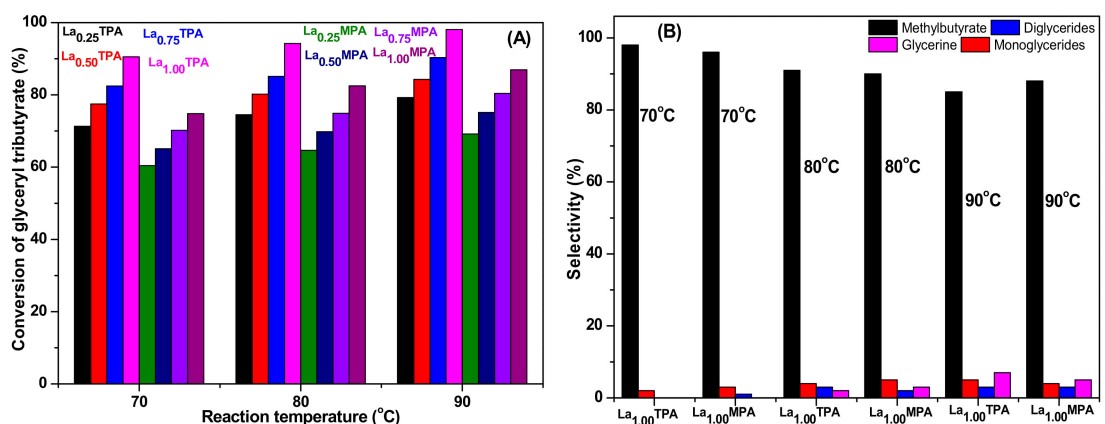

**Figure 6.** Influence of reaction temperature over (**A**) conversion of glyceryl tributyrate (**B**) selectivities of transesterification products.

The selectivities of the products observed are in accordance with different reaction steps of transesterification of glyceryl tributyrate, as shown in Equations (1)–(3). It was clear that initially the triglycerides converts into diglycerides, and subsequently into monoglycerides and glycerin. At low conversions, the selectivity to methyl butyrate is between 96% and 98% in the case of highly active $La_{1.00}TPA$ and $La_{1.00}MPA$ catalysts. The selectivity to methyl butyrate was slightly lowered at high reaction temperatures, due to the formation of more glycerin. Low selectivity to monoglyceride was observed due to the fact that monoglyceride has a tendency to convert into methyl butyrate and glycerin. It was observed that a maximum of 7% selectivity to glycerin occurred for $La_{1.0}TPA$ sample.

The influence of a reaction time over the conversion of glyceryl tributyrate and selectivity towards methyl butyrate for all the synthesized catalysts was studied at 90 °C and the results are shown in Figure 7. Initially, the glyceryl tributyrate conversion levels increased linearly until they reached 120 min and reached a peak at 140 min. A further increase of the reaction time has not affected the conversion levels, indicating that the equilibrium was established at 120 min. This is mainly due to the

fact that the transesterification continues via a stepwise reaction of the triglycerides, which led to the production of intermediate diglycerides and monoglycerides, which finally transformed into biodiesel and glycerin.

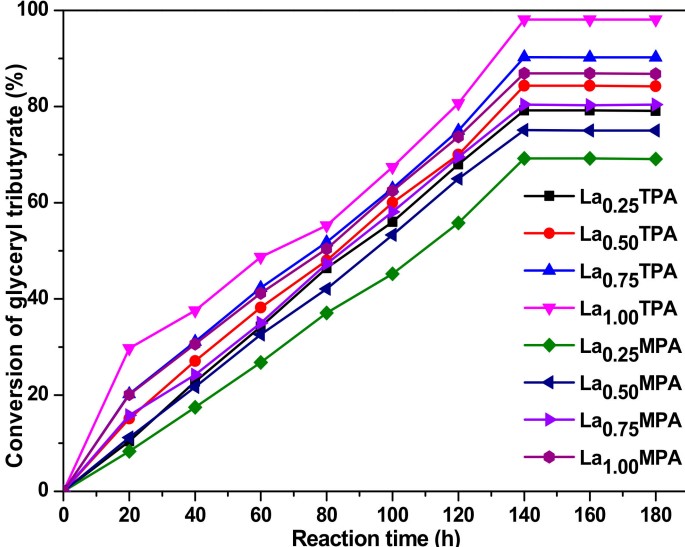

**Figure 7.** Influence of the reaction time on the conversion of glyceryl tributyrate over $La_xTPA$ and $La_xMPA$ catalysts.

As observed in the FT-IR spectral analysis of pyrrole adsorbed $La_xHPA$ samples, any increase in the $La^{3+}$ concentration (from 0.25 to 1.00) resulted in an increase in the number of basic sites of $La_xHPA$ samples, which subsequently influenced the transesterification activity. Figures 6 and 7 indicate that an increase of the $La^{3+}$ ion composition from 0.25 to 1.00 led to continuous enhancing of the activity, as the highest activity was found in the case of the catalyst that contained a $La^{3+}$ composition of 1.00. However, we have not increased the $La^{3+}$ composition beyond the stoichiometry of Keggin ion, due to the fact that incorporation of more counter cations could decompose the Keggin ion structure to form its oxides. The observed results are in accordance with the previous findings that basic sites of catalysts play a crucial role in transesterification. Among the synthesized catalysts, $La_{1.00}TPA$ was found to exhibit the highest catalytic activity towards the transesterification under optimized reaction conditions. The superior performance of the $La_{1.0}TPA$ sample could be related to its superior physico-chemical characteristics such as high surface area and a large number of basic sites per unit surface area.

The role of the methanol to glyceryl tributyrate molar ratio on the transesterification activity of the most active $La_{1.00}TPA$ catalyst was also studied (Figure S3). The transesterification reactions were conducted by changing the methanol to glyceryl tributyrate molar ratios of 3:1, 6:1, 9:1, and 12:1 for 1 h. The conversion of glyceryl tributyrate increased from 24% to 98% after increasing the methanol to glyceryl tributyrate ratio from 3:1 to 18:1. This result is consistent with a previous observation that the utilization of large molar ratios of methanol to glyceryl tributyrate is advantageous to obtain higher methyl ester yields [4,5]. This mainly due to the fact that excess alcohol does enhance the rate of transesterification and also assists in desorption of product molecules from the surface of the catalyst.

Further, we also studied the effect of loading of catalyst into the reaction mixture (Figure S4). Several transesterification experiments were performed at 90 °C for 140 min using a methanol to glyceryl tributyrate 12:1 molar ratio in the presence of different loadings of $La_{1.00}TPA$ catalyst (1, 3, 5, and 7 wt.%). It is clear that glyceryl tributyrate conversion was enhanced with an increase of the catalyst amount. A complete conversion of glyceryl tributyrate was observed in 150 min, when 5 wt.% catalyst loading was used, and longer reaction times were required when 1 and 3 wt.% loading was utilized. However, higher catalyst loading (greater than 5 wt.%) resulted in a minor diminution in



glyceryl tributyrate conversion. This is mainly because of the fact that the reaction mixture becomes viscous at higher catalyst loading, which could lower the rate of diffusion of reactants and products.

Having demonstrated that the La$_x$TPA samples possessed both acid and base sites, we performed simultaneous transesterification and esterification of palmitic acid reaction over La$_{1.00}$TPA catalyst to determine whether glyceryl tributyrate transesterification could take place in the presence of free fatty acid. This aspect is important as many biodiesel feedstocks consist of free fatty acids and the catalyst should able to work for one pot transesterification of triglycerides and esterification of free fatty acids.

The TOF values for La exchanged TPA and MPA samples along with bulk TPA and MPA catalysts are depicted in Table 3. The bulk TPA and MPA samples showed lower activity for transesterification of glyceryl tributyrate. Incorporation of low amounts of La (La$_{0.25}$ and La$_{0.50}$) in TPA or MPA resulted in an increase in activity for tributyrate transesterification. It is interesting to note that further La incorporation (La$_{0.75}$ and La$_{1.00}$) resulted in enhanced performance with the highest TOF obtained in the case of a La$_{1.00}$TPA sample. The highest transesterification rate of 12.8 mmol h$^{-1}$ g$^{-1}$ obtained in the La$_{1.00}$TPA sample coincides with the conventional basic MgO catalyst, which offered 15 mmol h$^{-1}$ g$^{-1}$ under similar reaction conditions. However, the La$_{1.00}$TPA catalyst underperformed compared to the previously reported Li-doped CaO and Mg-Al layered double hydroxides catalysts [37]. It is well reported that the transesterification activity of solid base catalysts depends on the strength of the basic sites. Formation of stronger and more basic sites in La$_{1.00}$TPA due to maximum La incorporation could be the responsible factor for the higher activity of La$_{1.00}$TPA. Moreover, HPAs are known to possess a very good resistance towards water and did not lose activity, in contrast to pure alkali and alkaline metal oxides.

**Table 3.** TOFs of transesterification of glyceryl tributyrate and palmitic acid esterification over the synthesized catalysts.

| Catalyst | TOF of Transesterification (mmol h$^{-1}$ g$^{-1}$ cat) | TOF of Esterification (mmol h$^{-1}$ g$^{-1}$ cat) |
|---|---|---|
| La$_{0.25}$TPA | 8.2 | 28 |
| La$_{0.50}$TPA | 9.6 | 24 |
| La$_{0.75}$TPA | 10.4 | 19 |
| La$_{1.00}$TPA | 12.8 | 13 |
| La$_{0.25}$MPA | 6.3 | 20 |
| La$_{0.50}$MPA | 7.5 | 16 |
| La$_{0.75}$MPA | 8.3 | 12 |
| La$_{1.00}$MPA | 9.8 | 8 |

It was also observed that the transesterification conversion was not affected due to the presence of palmitic acid, as the catalyst effectively converted the palmitic acid into its methyl ester in just 30 min of reaction time at a reaction temperature of 90 °C. It was reported that HPAs have a unique characteristic property where they could absorb the molecules into their bulk [38], which allows the reactions to happen in the bulk as well as on the surface [39]. The synthesized La$_x$TPA samples clearly possessed the Keggin HPA structure with balanced acid-base centers. Therefore, the reactant molecules could be migrating on the surface and bulk of HPA to undergo transesterification and/or esterification, with the subsequent products transported to the surface to desorb.

It is also known that some HPAs could dissolve in polar and non-solvents, therefore we performed a leaching test to determine whether any soluble HPAs were dissolved in the reactant components. As anticipated, the HPA samples which contained less La (La$_{0.25}$ and La$_{0.50}$) showed slight dissolution in hot methanol during the reaction. However, the samples which contained more La ions (La$_{0.75}$ and La$_{1.00}$) were stable during methanol reflux. Further, we studied the reusability of the synthesized La$_x$HPA catalysts. The transesterification of glyceryl tributyrate was carried out under studied reaction conditions. After the each cycle, catalyst was centrifuged and dried at 110 °C. The dried La$_{1.00}$TPA sample was used for five continual cycles under the identical reaction parameters described in the experimental section. The data obtained from recycle experiments was presented in Figure 8. It was

observed that the $La_{1.00}TPA$ sample offered a very similar conversion of glyceryl tributyrate without any significant loss. However, after the fourth cycle there was a slight decrease in the conversion of glyceryl tributyrate, this is possibly because of a loss of catalyst amount throughout the recovery and regeneration steps.

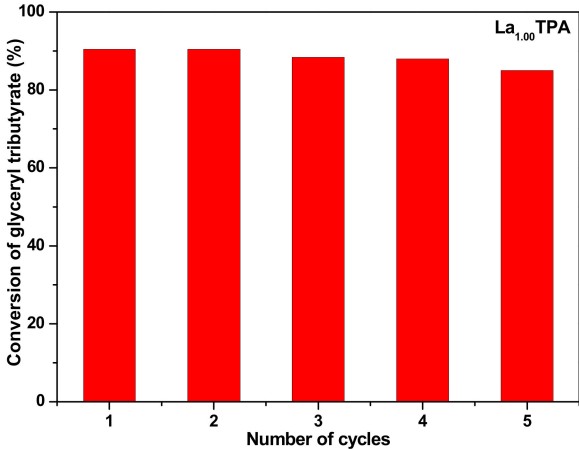

**Figure 8.** Reusability of $La_{1.00}TPA$ catalyst.

## 3. Experimental

### 3.1. Materials

Pure 12-tungstophosphoric acid $[H_3PW_{12}O_{40}\cdot nH_2O]$, 12-molybdophosphoric acid $[H_3PMo_{12}O_{40}\cdot nH_2O]$, lanthanum chloride $[LaCl_3]$, glyceryl tributyrate $[C_{15}H_{26}O_6]$, methanol $[CH_3OH]$ and hexyl ether $[C_{12}H_{26}O]$ were obtained from Aldrich and used as received.

### 3.2. Preparation of La Exchanged TPA and MPA Salts

A simple ion exchange method was adapted to synthesize La-modified HPA catalysts. To synthesize $La_xTPA$ (x = 0.25, 0.50, 0.75 and 1.00) powders, a calculated amount of TPA (dried at 110 °C) was dissolved in 100 mL of solvent (90 mL of ethanol and 10 mL of distilled water) by stirring to obtain a clear solution. The calculated amount of $LaCl_3$ $nH_2O$ corresponding to La ratio was added to 50 mL of solvent under stirring. A white precipitate was obtained by dropwise addition of $LaCl_3$ solution to TPA solution and the formed precipitate was centrifugally separated and washed four times with distilled water. The obtained precipitate was dried at 100 °C for 6 h. The $La_xMPA$ powders were also synthesized by following the similar procedure. Finally, all the dried samples were thermally treated at 300 °C for 4 h. Bulk TPA and MPA samples were prepared by calcining the 12-tungstophosphoric acid and 12-molybdophosphoric acid samples in muffle furnace at 300 °C for 4 h.

### 3.3. Materials Characterization

The elemental composition of $La_xTPA$ and $La_xMPA$ materials was determined by using an ICP-AES (Optima 7300 DV, Perkin-Elmer) instrument (Waltham, MA, USA). Powder XRD patterns of samples were collected using a PANalytical XpertPro diffractometer (Almelo, The Netherlands). The crystallite sizes of synthesized samples were determined by applying the Scherer equation.

The DR UV-vis spectra of $La_xTPA$ and $La_xMPA$ materials were obtained using a Thermo- Scientific evolution spectrophotometer in the range of 200–800 nm. The SEM images of the samples were obtained using a JEOL Model JSM-6390LV microscope (Peabody, MA, USA). The FT-IR spectra of calcined materials were obtained using a Bruker vertex 70 FT-IR spectrometer. The acidic and basic characteristics of $La_xTPA$ and $La_xMPA$ samples were investigated by obtaining the FT-IR spectra of pyridine and pyrrole adsorbed samples, respectively [40]. A calculated amount of the catalyst

sample was placed in an electric oven at 100 °C under vacuum for 2 h to remove the physiosorbed air constituents, including moisture. Then, the sample was exposed to pyridine or pyrrole vapor in a sealed desiccator and was heated at 100 °C under vacuum for 2 h to remove physically adsorbed pyridine or pyrrole before analyzing the sample. The analysis time in the ATR-IR technique is less than two minutes and the extent of moisture adsorption on the catalyst surface was reduced. The FT-IR spectra of samples was assessed before and after probe molecules adsorption, and we subtracted the spectrum of the sample before adsorption from the spectrum of the sample after adsorption using software. The amounts of Bronsted and Lewis acid sites were calculated via integration of the area of the absorption bands showing the maximum values of intensity at 1450 cm$^{-1}$ and 1545 cm$^{-1}$, respectively. Integrated absorbance of each band was obtained using the relevant software by applying the corresponding extinction coefficient, which was normalized by the weight of the sample. The XPS spectra of synthesized materials were obtained by using a Kratos Axis Nova spectrometer. The textural properties of the samples were obtained by carrying out the N$_2$-physical adsorption experiments using a Quantachrome ASiQ unit (Pleasanton, CA, USA).

### *3.4. Transesterification of Glyceryl Tributyrate with Methanol*

The catalyst evaluation for glyceryl tributyrate transesterification was accomplished using two neck RB flasks fitted with a magnetic stirrer, condenser, and paraffin oil bath. Glyceryl tributyrate and methanol with an optimized concentration (1:12 mmol) were first placed in the RB flask along with hexyl ether (internal standard, 2.5 mmol). Then, the flask was heated using an oil bath to the desired reaction temperature, then a calculated amount of catalyst (5.0 wt.% relative to tributyrate weight) was added to initiate the reaction. The liquid samples were withdrawn periodically to determine the product distribution using Shimadzu GC17A gas chromatograph integrated with DB-1 capillary column and flame ionization detector.

## 4. Conclusions

A simple ion exchange technique was adapted to synthesize La exchanged TPA and MPA salts (La$_x$TPA and La$_x$MPA, $x$ = 0.25, 0.50, 0.75 and 1.00). The obtained samples have been utilized as catalysts for transesterification glyceryl tributyrate mixed with methanol to produce biodiesel. A detailed characterization of the synthesized materials suggested that La$_x$TPA and La$_x$MPA materials composed of Keggin structure with La situated in the secondary structure. The XPS reveals that most of the samples possessed two environments (W$^{6+}$/W$^{5+}$ or Mo$^{6+}$/Mo$^{5+}$), indicating existence of a surface layer of HPA with reduced W$^{5+}$ or Mo$^{5+}$ ions. The FTIR analysis of pyrrole adsorbed samples and N$_2$-physisorption results disclosed that number of basic sites were increased with an increase of La loadings from 0.25 to 1.00. The highest number of accessible surface basic sites was observed in the case of a La$_{1.00}$TPA sample. The highest glyceryl tributyrate conversion, 98% was obtained at 90 °C, the molar ratio of glyceryl tributyrate and methanol is 1:12, while the reaction time is 140 min. The La$_x$TPA and La$_x$MPA materials with $x$ > 0.25 did not show any leaching and were reused several times without substantial reductions in their performance. The synthesized La exchanged HPA samples can be utilized for one pot of transesterification glyceryl tributyrate, palmitic acid esterification, and reactions due to the presence of balanced acid and base sites.

**Supplementary Materials:** The following are available online at http://www.mdpi.com/2073-4344/9/12/979/s1, Figure S1: FESEM images of La$_x$TPA and La$_x$MPA samples., Figure S2: Nitrogen adsorption-desorption and pore size distribution patterns of samples, Figure S3: Influence of methanol and glyceryl tributyrate ratio on the catalytic performance of La1.00TPA catalyst Figure S4: Influence of catalyst amount on the catalytic performance of La$_{1.00}$TPA catalyst, Table S1: Quantification of acidic and basic sites presented in the catalysts.

**Author Contributions:** B.A.-S. carried out all the experiments, analyzed the data and the mainly responsible for writing-original draft of the paper; K.N. and Q.A.A. writing-review and editing; K.N. conceived the research project and designed the experiments; K.N. and Q.A.A. supervised and directed the research project.

**Funding:** This research was funded by the Deanship of Scientific Research (DSR) of King Abdulaziz University, Jeddah, Saudi Arabia (grant number DF-082-130-1441).

**Acknowledgments:** This work was supported by the Deanship of Scientific Research (DSR), King Abdulaziz University, Jeddah under grant number DF-082-130-1441. The authors, therefore, gratefully acknowledge DSR technical and financial support.

**Conflicts of Interest:** The authors declare no conflict of interest.

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
