# Peer review of "Lanthanum Exchanged Keggin Structured Heteropoly Compounds for Biodiesel Production"

_catalysts, doi:10.3390/catal9120979_

Round 1

Reviewer 1 Report

Please, find in attachment the Reviewer comments. 

Author Response

Responses to reviewer comments

Reviewer 1

Abstract

The authors are requested to define the following acronyms: XRD, FT-IR, SEM, XPS and HPA.

Response: We appreciate reviewer comment, we defined the acronyms for XRD, FT-IR, SEM, XPS and HPA in the abstract.

Introduction

The authors are requested to define the following acronyms: XRD, FT-IR, SEM, XPS and HPA among others.

Response: We defined the acronyms in the Introduction as well.

Page 2, line 49-50 “These properties are significant for any catalyst …………..thus improving their recycle potential [13]. Please, after the previous sentence add the following sentence: “It is worth mentioning that employing heterogeneous catalysts for biodiesel production based on HPAs is timely and extremely important, because can be used in industry [Reference: Miriam Fontalvo Gómez Boris Johnson Restrepo Torsten Stelzer Rodolfo J. Romañach, Process Analytical Chemistry and Nondestructive Analytical Methods: The Green Chemistry Approach for Reaction Monitoring, Control, and Analysis https://doi.org/10.1002/9783527628698.hgc142] for this greener fuels [Reference: Sarmento J. Mazivila, Trends of non-destructive analytical methods for identification of biodiesel feedstock in diesel-biodiesel blend according to European Commission Directive 2012/0288/EC and detecting diesel-biodiesel blend adulteration: A brief review, 180 (2018) 239-247. https://doi.org/10.1016/j.talanta.2017.12.057] with easy separation between the target (biodiesel) and byproducts (mainly glycerol)”. Zhao et al. [14] synthesized ……….

Response: Authors appreciate reviewer suggestion, we added the suggested sentences and references in the revised manuscript.

Results and discussion

Fourier transformation-infrared (FT-IR) spectroscopy, Ultraviolet-visible (UV-vis) absorption spectroscopy, Scanning electron microscopy (SEM), X-ray photoelectron spectroscopy (XPS), X-ray powder diffraction (XRPD) techniques provided information needed to full characterize the material synthetized.

Finally, I strongly recommend the publication of the present report, with few minor revisions.

Response: Authors appreciate reviewer’s time and suggestion, we are grateful to reviewer for recommending the manuscript with minor revisions.

Reviewer 2 Report

The authors used La-exchanged MPA and TPA for the transesterification of glyceryl tributyrate with MeOH. The La content was varied, where the compounds with 1 wt% La showed the best catalytic activity. The samples were comprehensively characterized by several methods the necessity of which is however questionable. On the other hand, the quality of acidity/basicity characterization is left to be desired.

Remarks:

The authors prepared the catalysts by a “simple ion exchange method”, however, for my opinion it is a simple precipitation. It is expected that the percentage of exchanged protons varies with the La content. But the presented data of metal contents in Table 2 do not provide molar ratios. These should be given. Normally, from XPS the metal content is given in atom%, not wt%.

How the FT-IR spectra were measured (KBr or ATR-IR)?

Concerning acidity/basicity measurements; it is correct that pyridine and pyrrole were used as probe molecules. However the procedure used for such experiments (see respective references) is completely different from that used in the present work. For such studies in situ measurements are applied to avoid contact with air and moisture before and after adsorption. Furthermore, by inspecting the spectra prior and after adsorption the effect of probe interaction with the sample can be better studied by inspecting subtracted spectra, in which the respective bands are clearly observable (see e.g. Ref. 16). This is not possible from the presented spectra in Fig. 7 (because these are “normal” not in situ spectra of the samples pretreated with pyridine and pyrrole, respectively. Insofar, the quantification of the acid and basic sites is not possible. But quantitative data are necessary to evaluate the properties of the catalysts, properly.

For the catalytic testing 5wt% catalyst was used - to which this is related?

The XRD patterns change by incorporation of La. Is this only a change of modification or are there additional phases formed? Respective discussion is missing. From which reflections the crystallite size was determined?

The comparison of the UV-vis spectra from their presentation in Fig. 3 is difficult. Comparing the edge energy values would be more significant.

The FESEM images (Fig. 4) and Fig. 5 can be shown in the SI.

In Table 2: the molar ratios of particularly La/W and La/Mo should be given.

Lines 257-261: The amount of B and L sites is discussed, but a quantitative analysis is missing!

The spectra shown in Fig. 7 are not suited (see comments above) to discuss the acidity/basicity of the studied catalysts.

The short cuts in Eq. 2-4 are not understandable; the used substrate was glyceryl tributyrate.

Lines 329-330: the BET surface areas are similar and the basic sites per unit surface area were not calculated and mentioned before. Therefor such conclusion is not acceptable.

Line 341: the amount of catalyst is given in wt% - to which this is related?

All conclusions related to the acidity/basicity are not confirmed by relevant experimental results (see comments above).

Author Response

Responses to reviewer’s comments

Reviewer 2

The authors used La-exchanged MPA and TPA for the transesterification of glyceryl tributyrate with MeOH. The La content was varied, where the compounds with 1 wt% La showed the best catalytic activity. The samples were comprehensively characterized by several methods the necessity of which is however questionable. On the other hand, the quality of acidity/basicity characterization is left to be desired.

Response: Authors appreciate reviewer’s time and effort to improve our manuscript. We strongly believed that comprehensive characterization (including acidity/basicity measurements) of samples is essential to understand the role of physico-chemical properties of samples on the catalytic activity of synthesized samples.

The authors prepared the catalysts by a “simple ion exchange method”, however, for my opinion it is a simple precipitation. It is expected that the percentage of exchanged protons varies with the La content. But the presented data of metal contents in Table 2 do not provide molar ratios. These should be given. Normally, from XPS the metal content is given in atom%, not wt%.

Response: Authors thanks reviewer’s comments. It was reported in the literature that the counter cations (such as H+, Na+ and NH4+) in the heteropoly compounds could be easily exchanged with other metal cations. For this reason, authors believe that ion change method is the appropriate name. The chemical composition values of the synthesized samples in Table 2 are modified as reviewer suggested. The metal content obtained from XPS analysis was provided in atom % in the revised Table 2.

How the FT-IR spectra were measured (KBr or ATR-IR)?

Response: The FT-IR spectra of the samples were measured with ATR-IR technique.

Concerning acidity/basicity measurements; it is correct that pyridine and pyrrole were used as probe molecules. However the procedure used for such experiments (see respective references) is completely different from that used in the present work. For such studies in situ measurements are applied to avoid contact with air and moisture before and after adsorption. Furthermore, by inspecting the spectra prior and after adsorption the effect of probe interaction with the sample can be better studied by inspecting subtracted spectra, in which the respective bands are clearly observable (see e.g. Ref. 16). This is not possible from the presented spectra in Fig. 7 (because these are “normal” not in situ spectra of the samples pretreated with pyridine and pyrrole, respectively. Insofar, the quantification of the acid and basic sites is not possible. But quantitative data are necessary to evaluate the properties of the catalysts, properly.

Response: Authors thank reviewer for the comment. Authors apologize for not providing complete description of the adapted methodology for the FT-IR measurements for acidity and basicity determination. We agree with the reviewer’s opinion that in situ measurements are generally used to avoid contact with air and moisture before and after pyridine or pyrrole adsorption. Unfortunately, we don’t have the in situ cell available at present to carry out the measurements. For this reason, the catalyst sample was placed in an electric oven at 100 oC under vacuum for 2 hours to remove the physisorbed air constituents including moisture. Then, the sample was exposed to pyridine or pyrrole vapor in a sealed desiccator and then it was heated at 100 oC under vacuum for 2 hours to remove physically adsorbed pyridine or pyrrole before analyzing the sample. The analysis time in ATR-IR technique is less than two minutes and the extent of moisture adsorption on the catalyst surface is very less.

We indeed obtained the IR spectra of samples before and after probe molecules adsorption and we subtracted the spectrum of sample before adsorption from the spectrum of sample after adsorption using software.

The amounts of Bronsted and Lewis acid sites were calculated via integration of the area of the absorption bands showing the maximum values of intensity at 1450 cm-1 and 1545 cm-1, respectively. Integrated absorbance of each band was obtained using the software by applying the corresponding extinction coefficient and normalized by the weight of the samples.

For these reasons, authors argue that the acidity/basicity results presented in the manuscript are relevant.

For the catalytic testing 5wt% catalyst was used - to which this is related?

Response: Authors appreciate reviewer question, we used 5wt% of catalyst for activity tests. We chose 5 wt%, because our preliminary test results (effect of catalyst weight on conversion of tributyrate) indicated that 5wt% is optimum. We have not presented all the results in the manuscript by considering the length of the manuscript. The 5 wt% was calculated by considering the weight of the triglyceride (tributyrate) as 100%.

The XRD patterns change by incorporation of La. Is this only a change of modification or are there additional phases formed? Respective discussion is missing. From which reflections the crystallite size was determined?

Response: Authors thank reviewer for pointing out this important aspect. Authors performed a systematic characterization to study the structural changes due to incorporation of La. The XRD patterns showed clear indication for the formation of LaxHPA. There is a clear possibility for formation of mixed oxide phases (Mo-La-P, W-La-P) with amorphous nature. This argument was supported by the observation that the XRD patterns of LaxMPA samples are composed of few broad humps due to unknown amorphous material, which is not observed in case of LaxTPA samples. This is probably due to the fact that MPA thermal stability is lower than TPA and a slight amount of Keggin compound was decomposed into oxides in case LaxMPA samples.

Interestingly, the XPS results indicated the presence of two types of surface W and Mo species in case of samples with high La amount. These observations and the discussion was already included in the revised manuscript. The most intense XRD reflections for bulk TPA, MPA and La exchanged salts were used to determine the crystallite size.

The comparison of the UV-vis spectra from their presentation in Fig. 3 is difficult. Comparing the edge energy values would be more significant.

Response: Authors appreciate reviewer suggestion, we calculated the edge energy values and made the comparison in the revised manuscript.

The FESEM images (Fig. 4) and Fig. 5 can be shown in the SI.

Response: As reviewer suggested me moved the FESEM images (Fig.4) and Nitrogen adsorption-desorption and pore size distribution patterns of samples (Fig. 5) to SI.

In Table 2: the molar ratios of particularly La/W and La/Mo should be given.

Response: Authors appreciate reviewer suggestion, as reviewer indicated La/W and La/Mo atom ratios are provided in the Table 2 of revised manuscript.

Lines 257-261: The amount of B and L sites is discussed, but a quantitative analysis is missing!

Response: Authors apologize for the error, we added the detailed experimental methodology in the experimental section of the revised manuscript. The amounts of Bronsted and Lewis acid sites were calculated via integration of the area of the absorption bands showing the maximum values of intensity at 1450 cm-1 and 1545 cm-1, respectively. Integrated absorbance of each band was obtained using the software by applying the corresponding extinction coefficient and normalized by the weight of the sample.

The spectra shown in Fig. 7 are not suited (see comments above) to discuss the acidity/basicity of the studied catalysts.

Response: Authors thank reviewer comment. As previously described we adapted methodology which is widely accepted to determine the acidity/basicity measurements. Authors agree that there is a possibility for an error, however we believe that it is with in the acceptable limit.

The short cuts in Eq. 2-4 are not understandable; the used substrate was glyceryl tributyrate.

Response: Authors appreciate reviewer suggestion, we removed the abbreviations in Eq.2-4 of the revised manuscript. The glyceryl tributyrate is a well-known triglyceride, for that reason we abbreviated it as TG.

Lines 329-330: the BET surface areas are similar and the basic sites per unit surface area were not calculated and mentioned before. Therefor such conclusion is not acceptable.

Response: Authors appreciate noticing the error and we apologized for it. We calculated the basic sites per unit surface area values and presented in the supporting information of revised manuscript. In fact La1.0TPA sample possessed large number of basic sites per unit surface area.

Line 341: the amount of catalyst is given in wt% - to which this is related?

Response: Authors appreciate reviewer question, The 5 wt% was calculated by considering the weight of the triglyceride (tributyrate) as 100%.

All conclusions related to the acidity/basicity are not confirmed by relevant experimental results (see comments above).

Response: Authors provided a detailed experimental methodology adapted to determine the acidity/basicity of catalysts. And the procedure we followed is widely accepted, therefore the conclusions made in the manuscript are relevant.

Round 2

Reviewer 2 Report

The article is accepted in the revised form.